# Urban DAS Data Processing and Its Preliminary Application to City Traffic Monitoring

**DOI:** 10.3390/s22249976

**Published:** 2022-12-18

**Authors:** Hang Wang, Yunfeng Chen, Rui Min, Yangkang Chen

**Affiliations:** 1Key Laboratory of Geoscience Big Data and Deep Resource of Zhejiang Province, School of Earth Sciences, Zhejiang University, Hangzhou 310027, China; 2Bureau of Economic Geology, John A. and Katherine G. Jackson School of Geosciences, The University of Texas at Austin, Austin, TX 78712, USA

**Keywords:** urban seismic, distributed acoustic sensing (DAS), traffic monitoring, seismic signal processing

## Abstract

Distributed acoustic sensing (DAS) is an emerging technology for recording vibration signals via the optical fibers buried in subsurface conduits. Its relatively easy-to-deploy and high spatial and temporal sampling characteristics make DAS an appealing tool to record seismic wavefields at higher quantity and quality than traditional geophones. Considering that the usage of optical fibers in the urban environment has drawn relatively less attention aside from its functionality as a telecommunication cable, we examine its ability to record seismic signals and investigate its preliminary application in city traffic monitoring. To solve the problems that DAS signals are prone to a variety of environmental noise and are generally of weak amplitude compared to noise, we propose a fast workflow for real-time DAS data processing, which can enhance the detection of regular car signals and suppress the other components. We conduct a DAS experiment in Hangzhou, China, a typical metropolitan area that can provide us with a rich data library to validate our DAS data-processing workflow. The well-processed data enable us to extract their slope and coherency attributes that can provide an estimate of real traffic situations. The one-minute (with video validations) and 24 h statistics of these attributes show that the speed and volume of car flow are well correlated demonstrates the robustness of the proposed data processing workflow and great potential of DAS for city traffic monitoring with high precision and convenience. However, challenges also exist in view that all the attributes are statistically analyzed based on the behaviors of a large number of cars, which is meaningful but lacking in precision. Therefore, we suggest developing more quantitative processing and analyzing methods to provide precise information on individual cars in future works.

## 1. Introduction

The increase in the global population has sped up the process of social urbanization. The number of people living in the city is expected to reach 6.7 billion in 2050 according to the World Social Report 2020 from the United Nations. The management of information and resources has become a major challenge in the rapidly expanding urban environment and has motivated the government to seek effective approaches to city management and sustainable development. Building a smart city is one of the key solutions to address these urban challenges and optimize the use of information and limited resources. Sensing technology plays a pivotal role in designing the smart city ecosystem that contains several essential components, including data collection, processing, communication, and action [1]. DAS is an emerging sensing technology that has been developing rapidly in geophysics. The DAS system contains an interrogator unit that sends coherent laser pulses into the fiber-optic cable and uses the optical phase shift of the back-scattered light to measure the small change in the length of fiber (strain or strain rate) in response to acoustic vibrations [2,3]. This technique turns a fiber-optic cable into tens of thousands of sensors that provide almost continuous (meters apart) sampling of the urban environment. Figure 1a,b are simple schematics about how DAS works. In Figure 1a, when the length of the fiber is unchanged, it corresponds to a stable phase sequence of back-scattered light. When there is a disturbance occurring on the fiber (location A), the optical path behind it will be changed. As a result, the corresponding phases are also different (see the green curves). By calculating the phase differences, we can find where the disturbance occurs. One of the most attractive features of DAS application in metropolitan areas is that the existing telecommunication infrastructure has already formed a dense network of fiber-optic cables. Hence, the DAS technique has the great potential of becoming an effective low-cost tool for real-time, long-distance, and large-scale sensing of the city.

The utilization of DAS in urban environments is an emerging field in geophysical applications. The pioneering investigations of urban DAS were conducted on the campus of Stanford University using a 2.45 km long fiber-optic cable placed in the telecommunication conduits [4]. Later studies were conducted in a more urban-like environment, including Palo Alto, CA [5] and Pasadena, CA [6,7,8] and Sacramento, CA [9] and Perth, Western Australia [10]. The applications of urban DAS have focused on aspects such as near-surface imaging [4,11,12] and monitoring [13,14], city traffic tracking [5,8], earthquake detection [15,16] and identification of other characteristic signals [6,17,18] These earlier studies have demonstrated a number of promising usages of DAS in the urban environment. In this study, we acquire the DAS data in the metropolitan area of Hangzhou, China, a city with over 12 million population. Our study scales up the current practice of DAS acquisition to the city scale. This offers a unique opportunity to examine the potential and challenge of DAS in a more typical urban environment.

The high complexity and large volume nature of the urban DAS data require developing an efficient analysis workflow to better utilize the data and exploit the rich information of the urban environment. In this work, we conduct a preliminary study of the DAS data processing strategy and its potential application. We focus on the most abundant signals in DAS recordings, which are traffic signals from moving vehicles. Several earlier studies have adopted various processing strategies to extract urban traffic signals. Liu et al. [19] used an improved wavelet threshold and a dual-threshold algorithm to detect traffic flow. Van den Ende et al. [20] used a deep-learning-based deconvolution approach to improve the temporal resolution and detection accuracy of car signals. Wiesmeyr et al. [21] employed the Hough transform borrowed from the image processing field to estimate vehicle flow and the average speed of large vehicles. Thulasiraman et al. [22] adopted clustering algorithms and Kalman filtering techniques in data mining and signal processing literature to identify and track vehicles. More recently, Wang et al. [8] adopted the 4th root slant stacking method to estimate the mean vehicle speed and volume for each ten-minute data segment. These studies highlight the opportunities and challenges of the DAS system in urban traffic monitoring. However, few of these mentioned works have focused on the data processing and analysis of a metropolitan city (more than ten million in population) that has more challenging data, which is the current research gap.

Compared to the other works mentioned above, the innovation of our paper lies in that (1) Hangzhou is a huge city with over 12 million population. This makes the data conditions extremely complex, maybe more complex than any other work. Thus we design a unique processing workflow to deal with this kind of complex DAS data. (2) We propose to use two effective attributes (slope and coherency) to reflect the traffic situations, which have not been found in other papers.

This paper is organized as follows. We first briefly introduce the acquisition and characteristics of the urban DAS data in Hangzhou. Then we introduce in detail our processing workflow that integrates several computationally efficient algorithms to tackle various issues in urban DAS data. Next, we demonstrate two examples of real-time traffic tracking and daily status monitoring using the processed data. Finally, we discuss the potential applications of urban DAS based on the current work and point out several promising future research directions.

## 2. Hangzhou DAS Data Acquisition and the Dataset

We conducted two days of DAS data acquisition in the city of Hangzhou. The interrogator was mounted on a server cabinet in a local data center. We used the AP-sensing equipment that allows simultaneous recording of two channels for a maximum distance of 50 km. We selected two telecommunication fiber-optic cables that are deployed along the roadside that roughly follow an E-W direction (Figure 2). These two lines are specifically selected considering: (1) the quality of the fiber-optic cable with a relatively low (<0.3 dB/km) light loss rate; (2) a relatively long monitoring distance of over 20 km; and (3) the diverse urban environments that the cable sampled. The north line extends northward first and then eastward along the Yuhangtang Rd and terminates near the Hangzhou East railway station, with a total length of 24.8 km. The south line goes directly eastward along the Xixi Rd and Tianmushan Rd, both having several construction sites for subway and underground tunnels. The total length of the south line is 18.1 km. The DAS system recorded continuously at a time sampling interval of 0.0005 s (2000 Hz) and a spatial sampling interval of 2.45 m. The gauge length was set to 10 m during the entire acquisition period. A total of 7.5 TB of data were acquired during the DAS experiment.

The diverse urban environments generate vibrations from a variety of sources, such as traffic, construction site, and city infrastructure, leading to rich DAS signals with significant spatiotemporal variability. Figure 3a,b show two typical three-minute recordings of DAS data. The north line reveals constantly vibrating signals characterized by a band of strong vertical energy at several locations along the profile (e.g., near 20,000 m distance). These signals are either from intersections or bridges, the characteristic of which will be detailed later. We identify these prominent signals in DAS data (Figure 3a,b) and combine them with satellite imagery from Google Earth to distinguish different road segments along the fiber. Compared to the north line, the south line is dominated by traffic signals. The signal strength varies considerably along the profile, which mainly depends on the coupling condition of the cable to the surrounding medium, burial depth, and distance to signal sources.

Figure 3c shows a typical signal that is often seen as isolated energy groups with limited lateral extent (width). These signals are caused by strong vibration from the vehicle moving perpendicularly to the optical fiber at the intersection, with each energy peak representing a passing vehicle. Because the car is directly passing over the cable, the vibration of the optical fiber is significantly increased, the energy is focused on a few recording channels and is significantly stronger than the surrounding channels. This type of signal appears periodically due to the control of traffic lights. Figure 3d shows a typical bridge signal that maintains a high energy level throughout the day. The fiber cable is typically shallowly buried beneath the deck of the bridge and hence is more susceptible to the surrounding environment and can record more vibration energy. As a result, the energy of the signal on the bridge is always stronger than that of the road on either side. The two vertical lines represent the junction of the bridge deck with the connecting road. These two sites maintain the highest energy because they are the most unstable regions in this stretch of the road which create the most strong vibrations, so we can infer that their width corresponds to the length of the bridge. In urban DAS recordings, the most common signal is from moving vehicles (Figure 3e). Because of the limit of lateral sensing distance of the DAS system, the collected data mainly record vehicles moving on the same side as the optical fiber. The signal with a positive slope (e.g., right half in Figure 3e) corresponds to vehicles moving toward the end point of the fiber, whereas the negative slope indicates the opposite moving direction (i.e., toward the start point of the fiber). As the vehicle moves along the road, the optical fiber at different positions successively receives vibration signals generated by the vehicle, so the signal appears as a slash line, and the slope of the signal reflects the slowness of the vehicle. Typically, the amplitude of the vehicle signal could reflect the weight of the cars, with high amplitude and long duration signals (shown as wide event axis) generated by buses or trucks. We also identify strong signals near the end of the north line (Figure 3f), where the cable crosses a railway and is distributed parallel to the eastern and western sides of the railway. When the train passes by, the optical fiber on both sides of the railway picks up the vibration signals at the same time, so the image presents a symmetrical feature. The train is heavy-mass and fast-moving, thus corresponding to the strong-energy and low-slope features. The optical fiber is affected by vibration for a fixed period of time because of the fixed length of the train.

In this work, we select a part of Yuhangtang Rd as the study area (Figure 4) for a preliminary investigation of DAS in a typical urban environment. Figure 4a is the Google satellite imagery of the detailed road conditions. Figure 4b is the intersection where we placed the two monitoring cameras, which contain the above-ground roads and tunnel. We use the camera to monitor the traffic condition at the tunnel exit (Figure 4c). Figure 4c,d are the views of cameras 1 and 2, respectively. Figure 4e is the layout diagram of the monitored area in Figure 4c.

## 3. Methods

The extensive amount of data acquired by city-scale DAS systems requires developing an effective processing workflow with high computational efficiency for real-time applications. The diverse and complex urban environments lead to relatively low-quality DAS data, which imposes significant challenges to conventional data-processing workflows. Therefore, a processing workflow with a stable noise-suppressing ability is highly demanded. Here, we propose an integrated framework, containing several simple but fast processing units, to deal with these specific issues in the DAS data. We briefly explain the key point of each processing step that is necessary to understand the rationale of our proposed workflow.

### 3.1. Low-Pass Filter

Since the city roads are natural noise sources, raw DAS data recorded on the roadside contain a mixture of wavefields. Among them, our target wavefield is generated by the moving vehicles (car signals hereafter) that are mainly characterized by low-frequency energy (0–2 Hz) in the Fourier spectrum. To highlight the car signals and filter out most of the high-frequency noise, we apply a low-pass filter to the raw data. The frequency spectrum of a sample trace shows strong high-frequency content (Figure 5, black line), whereas the main energy of the car signals is focused in the low-frequency (0–2 Hz) band. Therefore, we design a low-pass filter (Figure 5, blue line) with an oblique edge to preserve the car signals (red line in Figure 5). The oblique edge decreases to 0 at around 2.5 Hz. Then, we multiply the raw spectrum (black line) by the low-pass filter (blue line). Since the frequency components higher than 2.5 Hz in the blue line are 0, the multiplication result (red line), also known as the car signal, will have the same zeros in the area higher than 2.5 Hz. This low-pass filter is sequentially applied to each trace (Figure 6a), and the resulting filtered section mostly contains the energy from moving vehicles (Figure 6b) with much of the high-frequency noise removed (Figure 6c).

### 3.2. Bad Trace Editing

The low-pass filtering step suppresses most high-frequency noise. However, there are still obvious bad traces existing in the filtered result. We define the bad traces as those with an anomalously large amplitude, though some of these traces contain physical signals (e.g., vibration ridge, see Figure 6b, between 8960 m and 9450 m). These parts of DAS data should be removed since a large amplitude may mask the car signals in the subsequent processing. We propose a simple and efficient approach to detect these bad traces. Considering their high amplitude feature, we calculate the average absolute amplitude of each trace (Figure 7) and set a threshold (red straight line). Traces with an amplitude value larger than the threshold are then removed from the 2D profile. The threshold value is obtained via a try-and-error strategy to make sure that it can reject as many bad traces as possible while creating minimal damage to the useful traces. Figure 8a,b show the processed DAS profile and its corresponding removed traces, respectively. This simple strategy is highly efficient and achieves robust performance in our tests.

### 3.3. Amplitude Scaling

Data quality is improved significantly after trace editing (see Figure 8a). However, only a few cars with relatively large amplitude signals are visible in the processed 2D section. To reveal the trajectory of the weak car signals hidden in the background noise, we apply an amplitude scaling strategy by clipping the data amplitude to 0.01, while data points with an amplitude smaller than 0.01 remain unchanged. Specifically, if the absolute value of a data point is smaller than 0.01, it can keep its value unchanged. However, if the absolute value is larger than 0.01, its original value will be changed to 0.01 multiplied by the sign of it. This procedure results in a more balanced energy distribution of the data (Figure 9).

### 3.4. Local F-K (Hard Thresholding) Filter

The amplitude-scaling process amplifies the car signals at the cost of raising the noise level. Thus, we adopt a local F-K filter with hard thresholding for further noise suppression. We define a local window with a center position (x, y) and window size (s×s). The distance between the centers of two neighboring local windows (i.e., sliding step *f*) is set to the same value in both *X* and *Y* directions. A local window comprises a basic processing unit, wherein the F-K transform is applied to filter out coefficients smaller than parameter τ in the 2D Fourier domain. This assumes that the noise is less regular and thus corresponds to the small values. By transforming the remaining coefficients back to the t−x domain, we complete the denoising process for a single unit. Finally, the denoising result of all processing units is integrated, and the overlapping areas are averaged. Figure 10a,b show the denoised data and removed noise sections, respectively.

### 3.5. Local F-K (Sector Cutting) Filter

Besides the aforementioned random noise, vertical and horizontal coherent noise are inherent characteristics of the DAS data. The former is mostly caused by poor coupling of the fiber cable or constantly vibrating sources, whereas the latter contains low-frequency surface waves propagating across the cable. We still utilize the local F-K framework but instead employ a different separating strategy, i.e., sector cutting. Since the vertical and horizontal events correspond to the extremely low and high apparent velocities, respectively, in the F-K domain, we select a sector area representing medium velocities and reject the rest of the spectrum energy. As a result, data in the time-space domain only contain normal events with reasonable slopes. Figure 11a,b demonstrate the denoising result and the removed coherent noise, respectively.

### 3.6. Curvelet Transform

To further enhance the filtering result and suppress the residual noise, we apply the curvelet thresholding operator to the result from the last step (see website http://www.curvelet.org/ for details about the curvelet filter, accessed on 1 October 2022). Based on the basic assumption of curvelet, the car signals with coherent events should correspond to the large value in the transform domain while the irregular residual noise is represented by the small coefficients. By clipping the small coefficients in the transformed domain, the quality of the result is further enhanced. Figure 12a,b illustrate the filtering result and the removed components.

### 3.7. Dip Filtering

The final step of the workflow is to separate the west- and east-heading cars by their event slopes since the two scenarios correspond to the negative and positive slopes, respectively. By applying a global F-K filter and dividing the transformed domain into positive and negative parts, we can obtain the separated wave fields. Figure 13a,b correspond to the west- and east-heading events.

Overall, the whole workflow can be summarized in Figure 14.

## 4. Results

### 4.1. Waveform Characteristics of Processing Results

The processed DAS section shows significantly improved SNR and reveals clear signals related to the moving vehicles (Figure 13a). These signals show strong variations in both amplitude and slope that can reflect the kinematic and kinetic information of cars. Specifically, the low-frequency dipping events in the DAS data are related to the subsurface deformation induced by the weight of the car [12,20]. The strength of this quasi-static deformation is controlled by both the distance and force exerted by the point load (i.e., cars). The corresponding particle velocity (i.e., the time derivative of the deformation) is determined by the speed of the car. Hence the strength of the car signals is ideally proportional to the weight of the car, and its slope values reflect the car speed. We noticed that the high-amplitude car signals (e.g., 7980–8450 m) are typically characterized by a wider waveform and steeper slope (note that the slope reflects slowness, not velocity) compared to low-amplitude ones. We attribute these signals to large, relatively slow-moving vehicles, such as buses and trucks that travel along the fiber. Figure 15 shows some typical large vehicles in panels 1–3 corresponding to the strong events. Additionally, the processing enables recovering some weak signals that are otherwise masked by the background noise, such as the small car in the fourth panel of Figure 15. However, notable amplitude variation exists in the processed DAS recordings, which is mainly caused by the installation conditions of fiber-optic cables. Overall, the processed DAS data section with clearer car signals enables us to extract useful attributes to estimate the traffic condition.

### 4.2. Speed and Volume Estimates of Traffic Flow

To quantitatively measure the car speed, we borrow the concept of local slope from the exploration seismology community calculated by the plane-wave destruction algorithm [23,24]. For a 2D data profile u containing several curved (or straight) car signals, assuming that each point has a dominant event slope *d*, the profile then corresponds to a slope field d varying along the time and space directions. This slope field allows us to design a destruction operator P(d) and destruct the waveform data u as
(1)r=P(d)u,
where r is the destruction residual. When the slope field d correctly reflects the real event dips, the residual r approaches zero. Thus, by minimizing the destruction residual, an optimized local slope map can be derived. Further, since the car velocity is closely related to the event slope, this slope map can then be converted into a velocity map via the following equation:(2)v=dx/dt/|d|,
where *v* is the car speed and *d* is the corresponding slope value. dt and dx are the temporal and spatial sampling intervals, respectively.

Figure 16a is a one-minute example of velocity distribution calculated based on the profile in Figure 13a. Because there are tunnel exits and crossroads (with traffic lights) between about 7900 and 8400 m (Figure 4, the cars passing the crossroads (monitored by cameras) generally reduce the speed, causing the blue (slow) zone in Figure 16a. Figure 16b is the corresponding screenshot at the camera location. The front cars are stopped at the traffic light so the cars behind exiting the tunnel or from the above-ground road all brake to slow down the vehicles.

Additionally, the volume of car flow is another important indicator reflecting traffic conditions. Before analyzing the volume of traffic flow, we made two prior assumptions: (1) cars should have stronger responses in the DAS data compared to the electrical bikes and bicycles since they have larger weight; (2) car signals are more coherent and consistent compared to the environmental noise. Thus, our task is turned into the identification of strong and coherent events in the DAS profile. We use a correlation-based multi-channel attribute cr to complete this task, which has also been used for extracting the first break with a strong linear feature in recent literature [25]. The calculation of this attribute cr can be expressed as follows:(3)$$\mathrm{cr}\left(i_{0},j_{0}\right)=\frac{1}{2l}\sum_{j=j_{0}-l,j\neq j_{0}}^{j_{0}+l}\max[\mathrm{corr}\left(x_{j_{0}},x_{j}\right)],$$
where $\left(i_{0},j_{0}\right)$ is the central index of a local window C with size (2w+1)×(2l+1) (i.e., it consists of (2l+1) channels and (2w+1) time points). $x_{j_{0}}$ stands for the central channel and $x_{j})$ represents other channels except for $x_{j_{0}}$ within the window. corr indicates their correlation function, and max means picking the maximum value in each correlation function. The final attribute value at point $\left(i_{0},j_{0}\right)$ is obtained by averaging all the 2l maximums. Repeating the above procedures at each point can output a 2D attribute map. As car events tend to have large amplitudes and high coherency, their corresponding locations can be effectively highlighted on the map.

Figure 17a is a one-minute example of a 2D attribute map also based on the profile in Figure 13a, which shows the clear car-flow distribution (note that, due to the fiber-coupling problems, some areas (8500–9500 m) can barely record vibrations). Since our camera is located at about 8350 m, we extract the attribute from the nearby channels (white rectangle) and plot these channels in Figure 17b. The corresponding road conditions from the monitoring camera are shown in Figure 17c–e. The red rectangle corresponds to the first monitoring picture that contains many west-heading cars and shows the highest coherency. Then, the black rectangle indicates a low-level coherency, and the corresponding monitoring picture shows no cars but only electrical bikes. At last, the coherency value shows a slight increase (blue rectangle), reflecting a new wave of traffic activities but with fewer cars than the first wave.

## 5. Discussion

### 5.1. Extracting Useful Information from the Spatiotemporal Variation of Attribute Maps

Based on the successful applications of the two attributes in the analyses of traffic situations mentioned above, we explore their spatiotemporal variation over a longer (24 h) period. A daily spatiotemporal distribution map of the correlation-based attribute is constructed by calculating the time average of each one-minute attribute map (see Figure 17a); this forms a row of the 2D daily attribute map (Figure 18) with a brighter color indicating a larger attribute value. The map demonstrates considerable variation in the coherency value in both spatial and temporal directions.

We first examine the temporal change in coherency values by averaging the daily variation of all channels in the 2D map. The resulting temporal trend shows a notable decline in traffic volume after 12:00, which is caused by the reduced number of vehicles after the morning rush hours. The second peak of traffic volume is observed between 18:00 and 23:00, corresponding to a long-lasting night rush hour. After midnight (00:00), the car density decreases rapidly and reaches the minimum around 4:00, after which the traffic level gradually recovers and surges again in the next morning rush hour at about 6:30 and peaks at around 10:00. This cycle of 24 h variation agrees well with the expected pattern of traffic flow in the monitoring area.

Besides the time-variation chart, we also average all-time slices to obtain a spatial variation chart (bottom panel of Figure 18). The large fluctuation of the coherency value mainly reflects the inherent characteristics of the DAS acquisition system. Specifically, the energy variation along the spatial direction is primarily determined by the cable installation conditions, such as burial depth and the coupling situation. Near the road intersections, the cables are usually buried deeply and thus record a minimum amount of vibration energy. This causes the low-energy vertical strips (e.g., channels near 8470 and 8960 m) on the 2D map. In the normal road sections, the coupling situations are generally reasonable, and the cables are shallowly buried, thus recording relatively higher energy. Near some special facilities, such as bridges and tunnel exits, the energy level is extremely high. These phenomena may be caused by (1) the shallow burial depth of the cable; (2) better coupling resulting from a generally harder pavement material of the road surface; and (3) the amplification of the vibration energy by the infrastructures. We suggest that the spatially varying coherency provides an estimate of the average daily energy level recorded by the fiber-optic cable, whereas the fluctuation superimposed on this trend (i.e., temporal variation) is caused by the passing vehicles. In other words, the comparison of the traffic flow at various positions may not be trivial without properly taking installation conditions into account.

Figure 19 shows the corresponding 24 h speed attribute. The average temporal speed variation shows a clear reversal trend compared to that of the correlation-based attribute. This phenomenon can be well-explained by the fact that the denser traffic flow in rush hours commonly causes the slowing down of cars and even traffic jams. While at midnight (23:00–5:00), the vehicles are generally moving at a speed close to, or slightly above, the speed limit due to a clear road condition.

Similar to the correlation-based attribute, the spatial variation of the speed attribute primarily reflects the intrinsic features associated with infrastructures. For example, two typical patterns are found near the intersections. First, if an intersection contains large-volume two-way traffic flows (i.e., a busy intersection with a large amount of north-southbound cars), the crossing cars can generate short horizontal events on the DAS data, which in turn cause a large value on the speed map, i.e., the yellow strip at around 8470 m. Second, near the small intersections (7980 m and 9700 m) or T-shape intersections (7980 m) with primarily east–westbound cars, the traffic control leads to the slow down of the vehicle, and thus causes low-speed strips. Additionally, the exit and entrance of the tunnel are also factors that reduce the speed of the car according to traffic regulations.

### 5.2. Potential of Urban DAS on Real-Time Traffic Monitoring

The presented processing workflow is highly efficient and thus is suitable for real-time application. Currently, without parallel computing of all procedures, processing a one-minute 2D DAS gather takes less than a minute. Specifically, the low-pass filtering takes the longest time, 36.9 s. Considering the possible frequency aliasing, the low-pass filtering should be taken before the downsampling to ensure no low-frequency contamination of the aliased artifacts. After low-pass filtering, the downsampling step (120,000×1225→300×1225) will make the data size 400 times smaller, thereby making all other filers very fast. Then, trace editing, amplitude clipping, local hard-thresholding FK filtering, local sector-cutting FK filtering, curvelet, and dip filtering take 0.002 s, 0.005 s, 2.4 s, 4.7 s, 0.35 s, and 0.09 s, respectively. The overall processing time is around 53.9 s. With simple parallel low-pass filtering, the overall processing time could be easily decreased to within 10 s (assuming 12 parallel threads), which is sufficiently fast for real-time traffic monitoring.

### 5.3. Enhancing the Existing Monitoring Capability via Advanced Signal Processing Techniques

The main challenge of effectively utilizing the DAS data is the very strong environmental noise, as illustrated in many examples mentioned previously. As a pioneering work on leveraging DAS for traffic monitoring in representative urban areas (e.g., Hangzhou), we apply the most basic signal processing methods, e.g., bandpass filtering, FK, to obtain a reasonably high-quality dataset. We use the curvelet method [26] to smooth the coherent signals at the risk of over-smoothing. Due to the fast development in filtering algorithms in the reflection seismology, there are many advanced signal-processing methods that could further boost the SNR of the urban DAS datasets. One of the noteworthy signal-processing approaches is the damped rank reduction (DRR) method [27,28], which can better separate the spatially coherent and incoherent components with minimized damages on the energy of coherent signals. Another possible improvement is to apply the high-resolution linear Radon transform that was recently used by Ref. [29] for preconditioning the teleseismic wavefields. The high-resolution linear Radon transform could potentially improve the resolution of each linear event (i.e., representing an individual vehicle) and optimize the vehicle volume estimate. Due to inevitable damages or over-smoothing of the coherent traffic signals, there exist observable coherent leakage signals in the removed noise. Such leaked signals (mostly weak) [30] could affect the quantitative measure of the traffic volume and should be minimized. Several recently proposed approaches, such as the local orthogonalization methods [30,31] or residual dictionary learning methods [32,33], could potentially help retrieve those leaked signals and improve the amplitude fidelity of the traffic signals.

## 6. Conclusions

DAS is an emerging research topic in the seismological community. Compared to the traditional seismic acquisition instruments, a continuous fiber line enables extremely dense spatial sampling of the wavefield. Additionally, fiber-optic cables can be conveniently deployed in the subsurface as a whole sensing device, which can reduce a considerable amount of time costs. Especially in the city environment, optical fibers have been installed roadside in advance for telecommunication purposes. These preexisting infrastructures offer a network of fiber cables used for vibration sensing. In this paper, we collect a DAS dataset in a typical urban environment of Hangzhou, China, and explore its potential for monitoring traffic situations. Firstly, to separate the car signals from the complex wavefields, we design a robust processing workflow integrating several fast and effective modules. This well-designed workflow effectively enhances the traffic signals and filters out other vibration components. On the basis of the well-processed data, we calculate two statistical attributes to examine the relationship between the DAS signals and real traffic situations, including (1) slope that directly reflects the actual vehicle speed and (2) spatial coherency that approximates the volume of the traffic flow. Both of them are validated via the one-minute and 24 h datasets. Finally, the spatial variation of these attributes that reflects the installation conditions of fiber-optical cable is closely related to various types of infrastructures, such as intersections, bridges, and tunnels. Overall, the proposed processing workflow enables us to monitor real traffic situations and demonstrate promising potential in large-scale urban applications. As for the limitation of this work, we statistically analyze the behaviors of a large number of vehicles but ignore the information of the individual cars. This will be solved in future works.

## Figures and Tables

**Figure 1 sensors-22-09976-f001:**
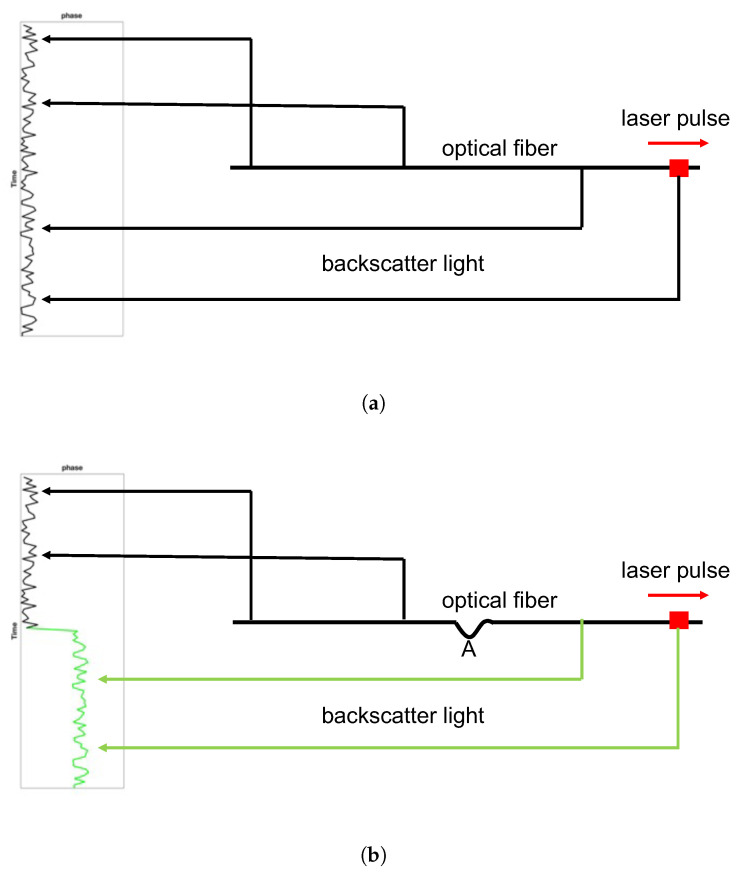
DAS schematic. (**a**) The situation of no disturbance occurring. (**b**) The situation with disturbance occurring on the fiber. The green color stands for the changed light paths and phases.

**Figure 2 sensors-22-09976-f002:**
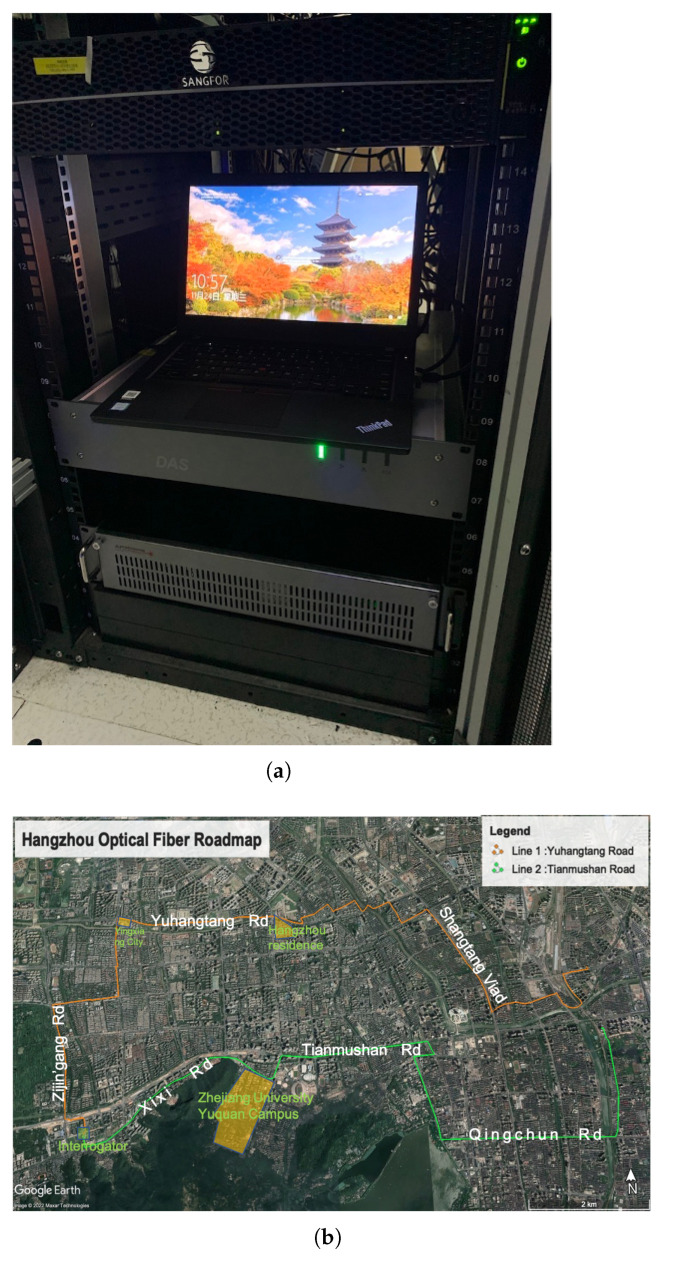
DAS data acquisition in Hangzhou. (**a**) AP-sensing interrogator unit. (**b**) Fiber-optic cable layout. The orange and green lines show the approximate location of the cables.

**Figure 3 sensors-22-09976-f003:**
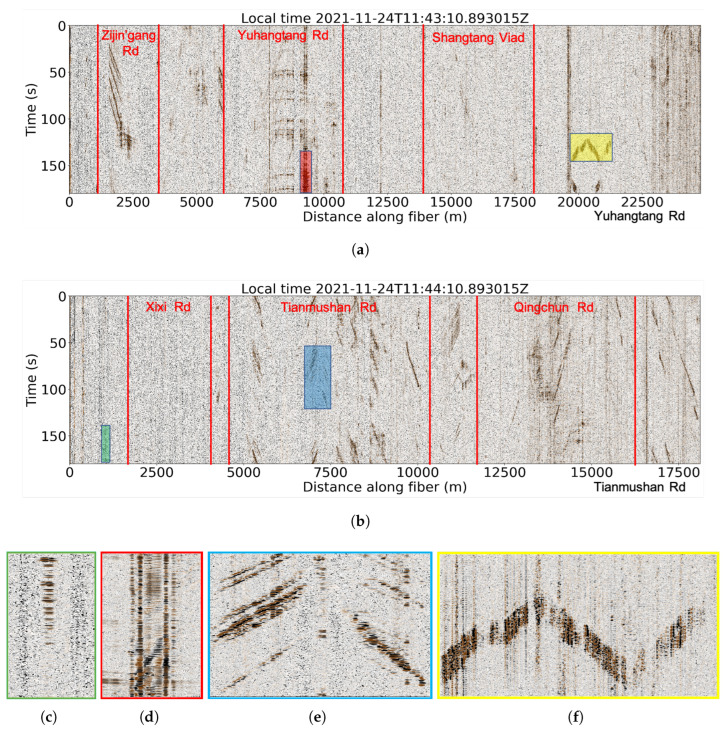
DAS data recordings from (**a**) north and (**b**) south lines. Characteristic urban DAS signals from (**c**) vehicles moving across optical fiber at the intersection, (**d**) bridge, (**e**) vehicles moving parallel to the optical fiber, and (**f**) trains.

**Figure 4 sensors-22-09976-f004:**
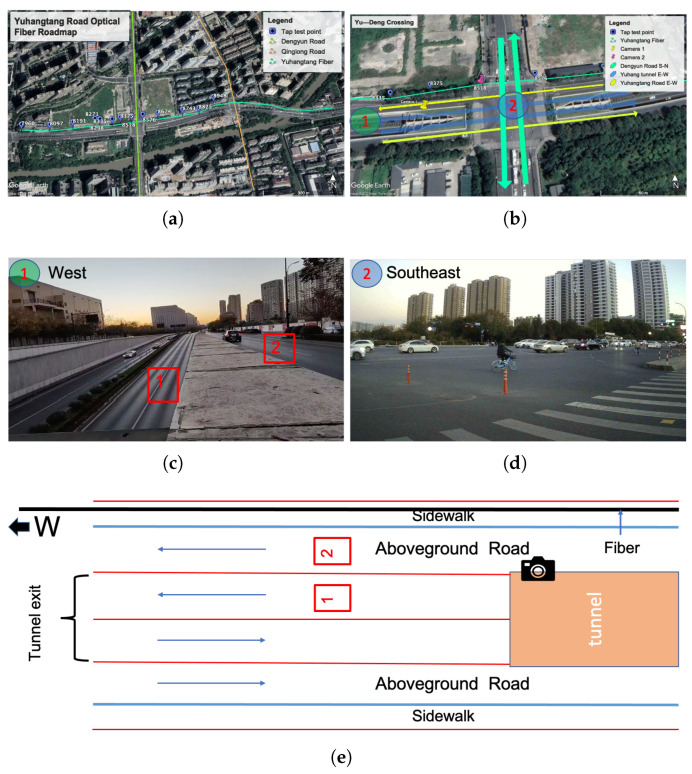
(**a**) Google satellite imagery showing the road section analyzed in detail in this study. (**b**) A zoom-in plot of the intersection that is monitored by two video cameras (yellow and red pins). The arrow indicates the traffic flow directions. (**c**) The tunnel exit is monitored by camera 1 facing the west direction. (**d**) The intersection is monitored by camera 2 facing the southeast direction. (**e**) A diagram illustrates the road layout of the monitored area of camera 1.

**Figure 5 sensors-22-09976-f005:**
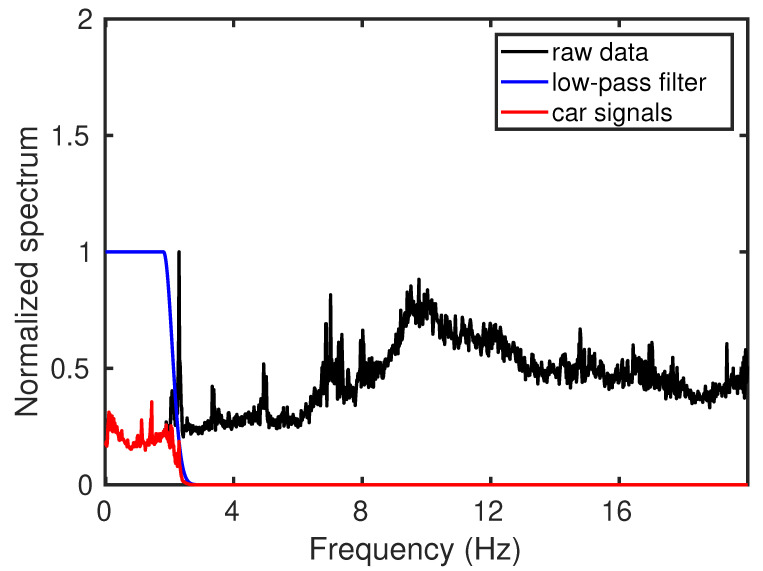
A diagram showing the frequency content of the urban DAS data and low-pass filtering process for car signal isolation. The black line indicates the amplitude spectrum of raw data. The blue line shows the transfer function of the low-pass filter. The red line shows the amplitude spectrum of filtered data that mostly contain car signals.

**Figure 6 sensors-22-09976-f006:**
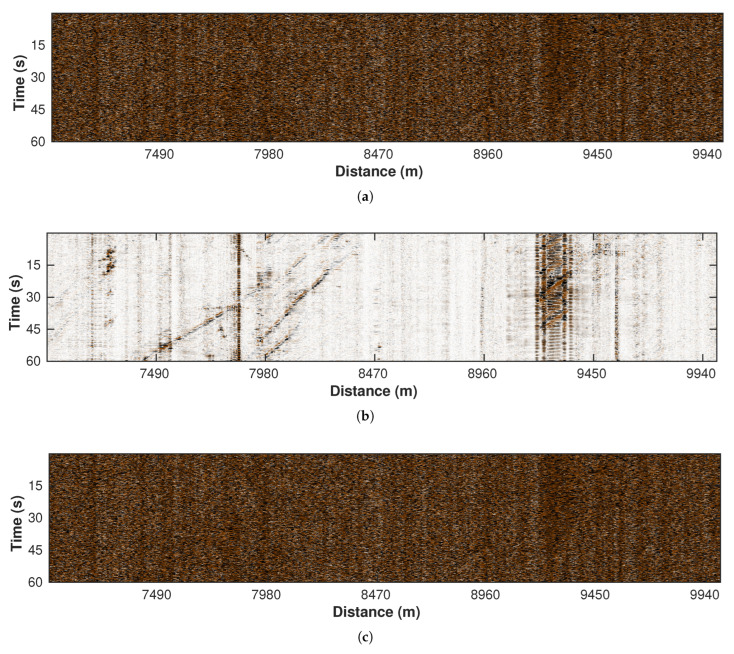
Processing results of low-pass filtering. (**a**) Original noisy data. (**b**) Low-pass filtered data. (**c**) Removed noise.

**Figure 7 sensors-22-09976-f007:**
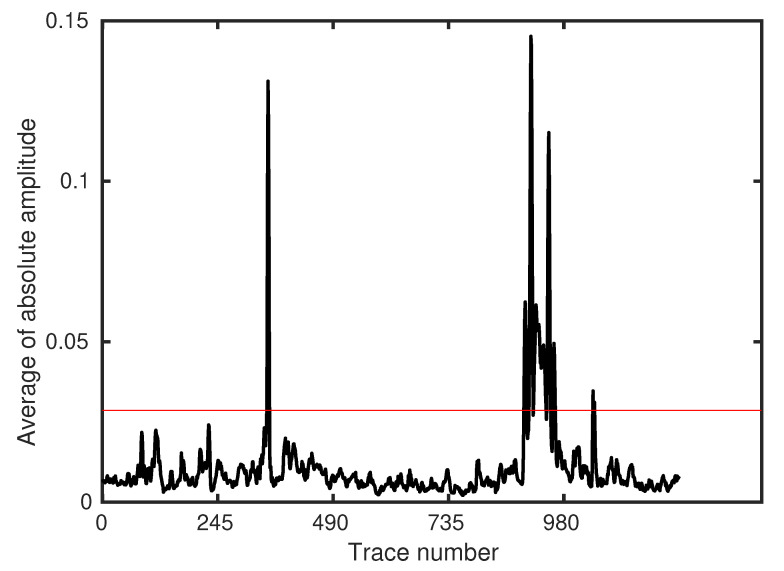
Average absolute amplitude of each trace in a DAS profile. The red straight line indicates the threshold above which the channels are considered bad traces and are removed from subsequent processing.

**Figure 8 sensors-22-09976-f008:**
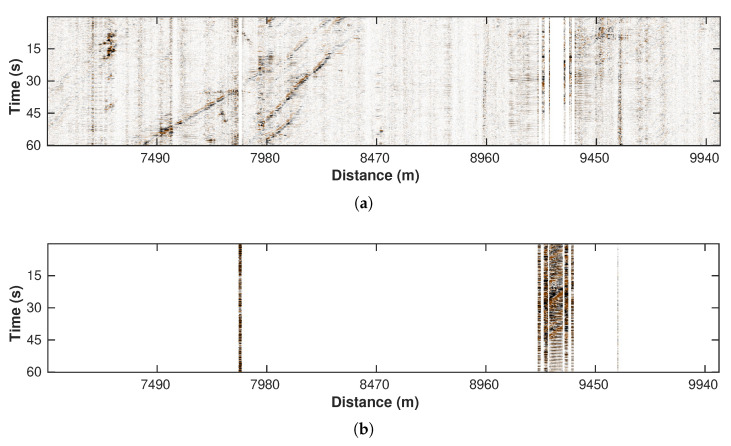
(**a**) Trace editing result. (**b**) Removed traces.

**Figure 9 sensors-22-09976-f009:**
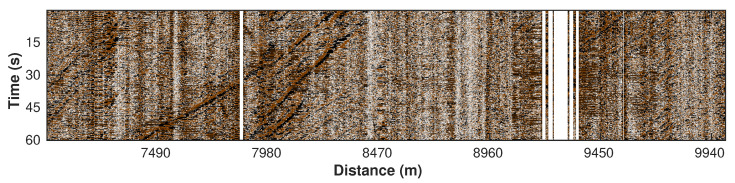
Amplitude scaling result. Data point with absolute value greater than threshold value is set to 0.01 or −0.01 (according to its sign).

**Figure 10 sensors-22-09976-f010:**
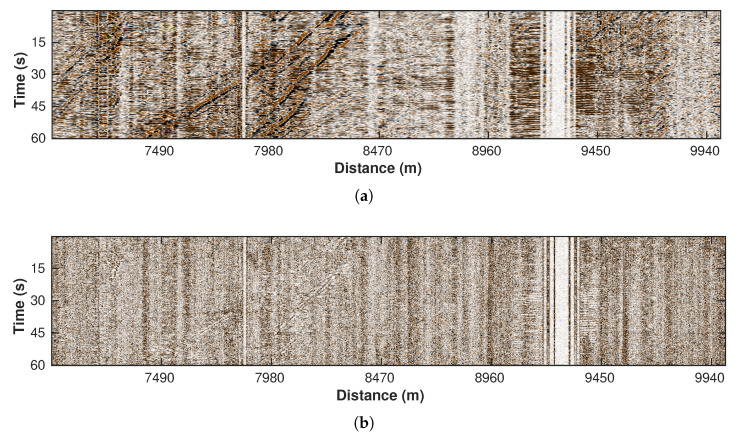
Processing results using local F-K filter with hard threshold. (**a**) Filtered data. (**b**) Removed noise section.

**Figure 11 sensors-22-09976-f011:**
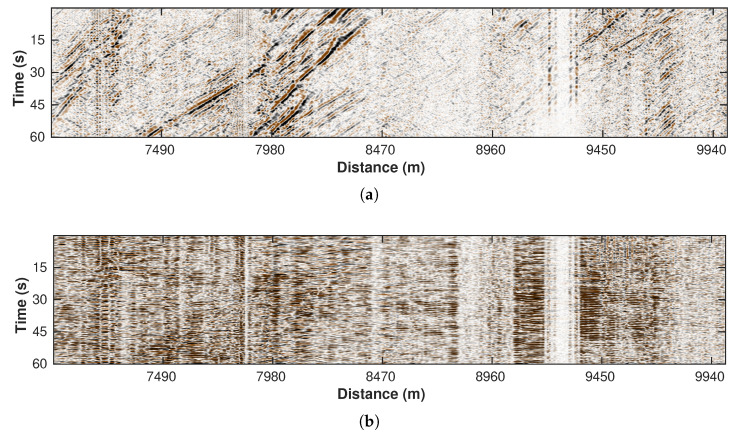
Processing results of local F-K filtering with sector cutting. (**a**) Filtered data. (**b**) Removed noise section.

**Figure 12 sensors-22-09976-f012:**
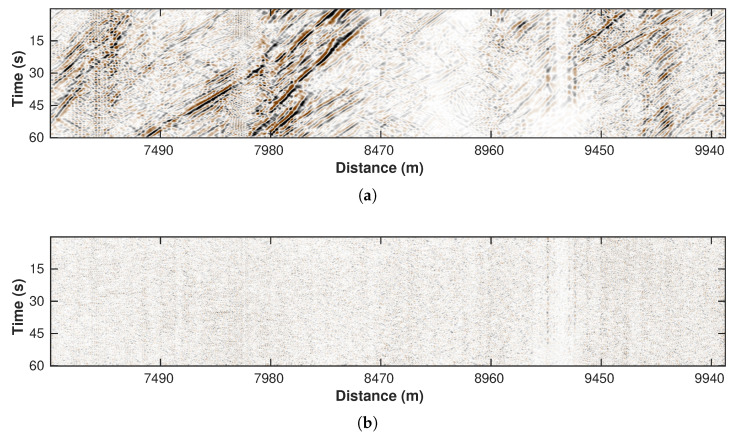
Processing results of curvelet transform. (**a**) Filtered data. (**b**) Removed noise section.

**Figure 13 sensors-22-09976-f013:**
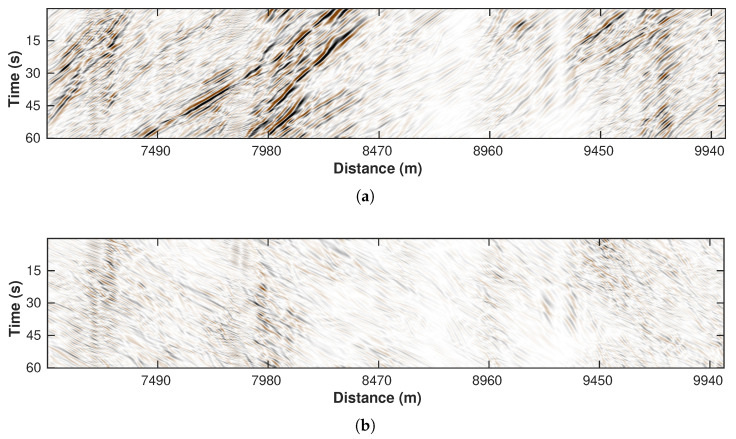
Processing results of dip filtering. (**a**) West-heading traffic signals. (**b**) East-heading traffic signals.

**Figure 14 sensors-22-09976-f014:**
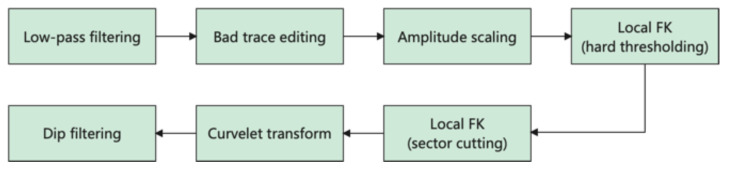
A flow chart showing the processing steps.

**Figure 15 sensors-22-09976-f015:**
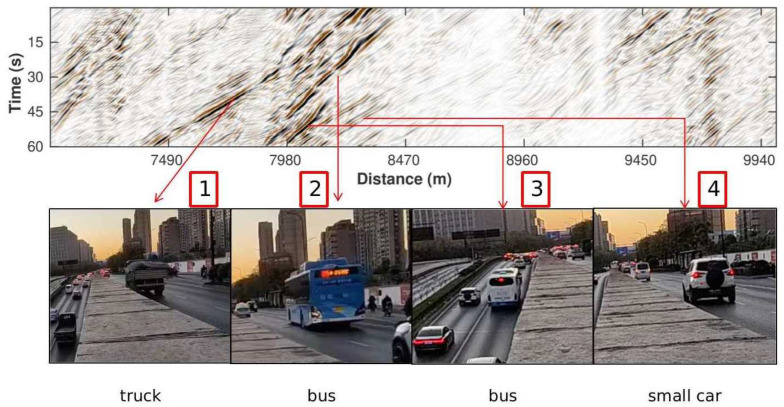
The processed west-heading events and their corresponding vehicles. Panels 1–3 show examples of large passing cars that induce strong signals, and the last panel shows an example of small car related to weak signals.

**Figure 16 sensors-22-09976-f016:**
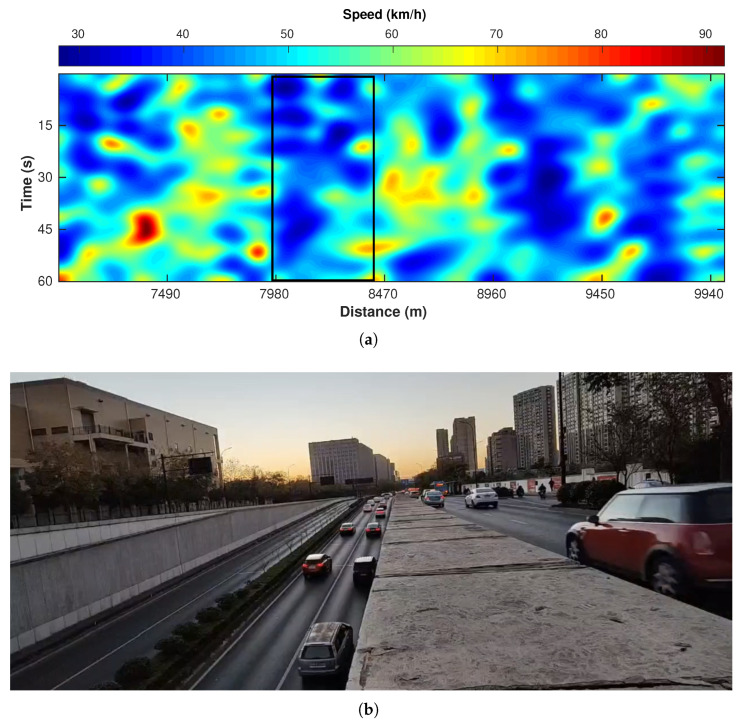
An example of traffic speed monitoring. (**a**) Spatiotemporal variation (one minute) in vehicle speed. The black rectangle highlight the zone with slow traffic. (**b**) A picture showing the slow down of cars, corresponding to the road segment between 7980 and 8400 m.

**Figure 17 sensors-22-09976-f017:**
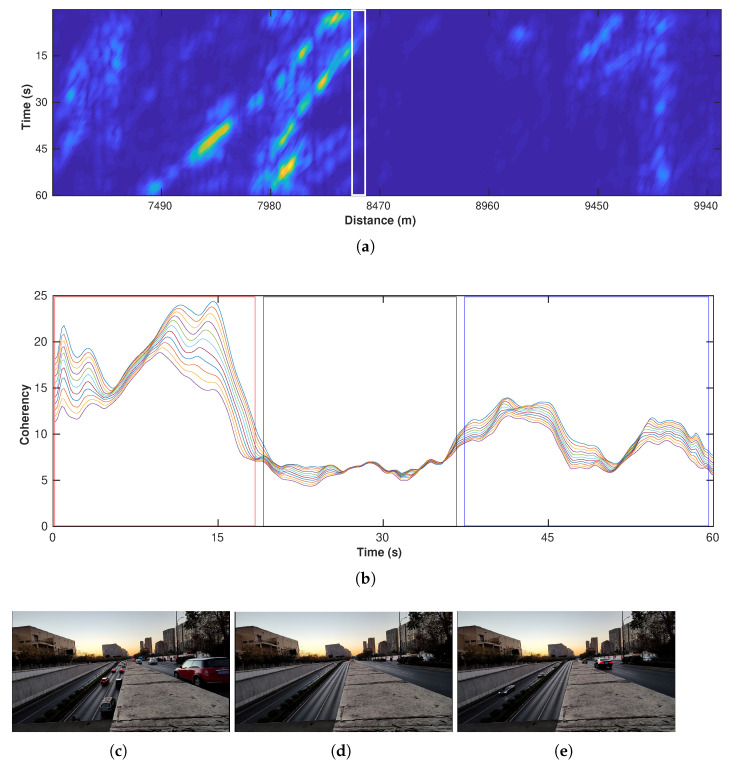
An example of traffic volume monitoring. (**a**) A 2D map of a one-minute correlation-based attribute. The white rectangle indicates the location of the 11 channels shown in (**b**). (**b**) The coherency values of selected channels near the camera. The red, black, and blue rectangles highlight the time periods with (**c**) many cars, (**d**) no cars, and (**e**) a few cars, respectively.

**Figure 18 sensors-22-09976-f018:**
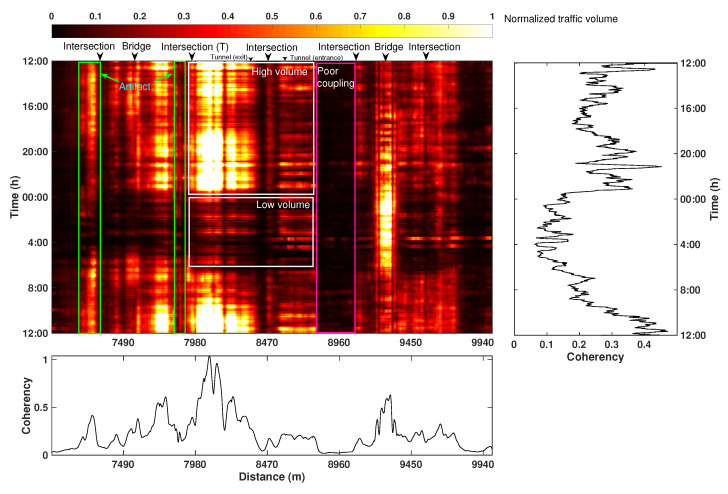
A 24-h spatiotemporal variation map of the correlation-based attribute. On the right side is the averaged result along the spatial direction. On the bottom is the averaged result along the temporal direction. Some typical landmarks causing these features are marked on the top of the 2D map.

**Figure 19 sensors-22-09976-f019:**
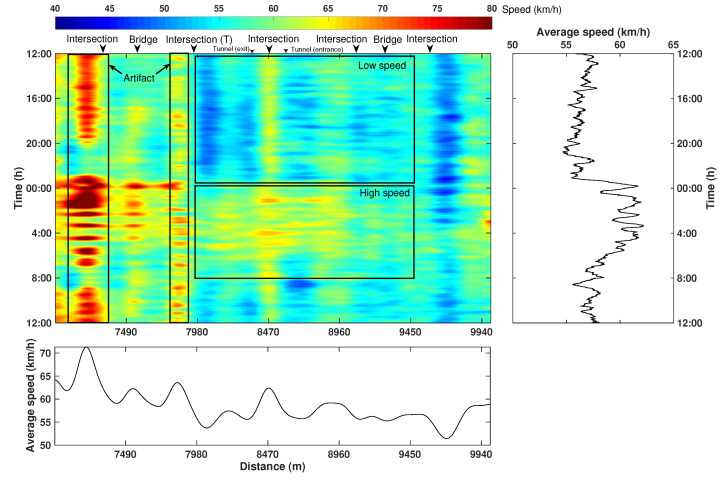
A 24 h spatial–temporal variation map of the speed attribute. On the right side is the averaged result along the spatial direction. On the bottom is the averaged result along the temporal direction.

## Data Availability

Data will be available upon request.

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
