# Peer review of "Urban DAS Data Processing and Its Preliminary Application to City Traffic Monitoring"

_sensors, 2022, doi:10.3390/s22249976_

Round 1
Reviewer 1 Report
The paper focuses on the application potential of DAS data for traffic monitoring. The research design is appropriate, and the methods are adequately described. The results are presented and illustrated well.
The paper is organized well, but the authors should make some improvements before publishing.
I recommend authors add one sentence in the Abstract to present the results, limitations of their work, plans for future work in the field.
The Introduction provide sufficient background. It shall be good if the authors state the research gap in the field.
I recommend authors to state the limitations of their study in the Conclusion.
Author Response
Please see the attached reply letter.

Reviewer 2 Report
DAS system is widely studied and widely used in earthquake, oil and gas industry, electric power and other fields. Signal processing is the focus of DAS research. In this paper, DAS is applied to urban traffic monitoring and a real-time DAS data processing method is proposed. It does not explain the details of the methods and signal processing. My advice is that a revision is necessary concerning the following.
1. The Abstract needs to be more concise, concise and clear.
2. In the Introduction, if the mechanism of DAS and the research content of this work can be shown in a schematic. It will be more acceptable to readers.
3. The Introduction is not a comprehensive review. The author should highlight the innovation of this paper, which is different from the previous research or has no such correlation. So the significance of the research is not obvious. It is suggested to further affirm the innovation of this work.
4. Figure 2 (c) ~ (f) lack detailed discussion. It does not explain the results and conclusions caused by this phenomenon.
5. Line 155, what is the low frequency? Or is it a range?
6. In Figure 4, the signal of low-pass filtering and the signal of vehicle are 0 at 2.5Hz. More explanation is needed.
7. What is the basis for setting the threshold (red line in Figure 6)?
8. Section 5.2, it is incorrect to say that a thing/event is robust without explanation.
9. How to consider SNR? Low-pass filtering is the most basic usage. Can a single low-pass filtering meet the requirements of this paper?
10. As a technical paper, the signal processing algorithm proposed in this paper needs more explanation.
11. Section 5.4. Future research should not be included in this paper. It should be updated in your future work.
12. Some of the used acronyms are not specified at first apparition, and the rest of the article only needs to use the specified letter. Please check it carefully. Such as DAS.
Author Response
Please see the attached reply letter.
